# Impact of severe mental illnesses on health-related quality of life among patients attending the Institute of Psychiatry, Rawalpindi from 2019 to 2021: A cross-sectional study

**Zarnain Umar** [ID]*, **Zona Tahir, Asad Nizami**

Institute of Psychiatry, Benazir Bhutto hospital, Rawalpindi, Pakistan

* zarnain.umar@hotmail.com

## Abstract

### Background

Severe mental illnesses have huge impact on health-related quality of life. They contribute to significant morbidity in terms of number of number of years of life lost in form DALYS (disability adjusted life years) and shorter life expectancy and early mortality. There is limited evidence on their burden in low- and middle-income countries like Pakistan.

### Objective

To determine the health-related quality of life in patients suffering from severe mental illness (schizophrenia, depression, and bipolar affective disorder) and demographic factors associated with poor health related quality of life in these patients.

### Methodology

This was descriptive cross sectional, using retrospective record view of data. Study was done under IMPACT (Improving Mental And Physical health Together) Program, which conducted a multi-morbidity survey conducted at institute of psychiatry, Benazir Bhutto hospital, Rawalpindi, using EQ 5d 5l (EURO QOL 5D5L) questionnaire having both subjective (EQVAS) and objective domains.

### Results

The study included 922 SMI patients, of whom 555 participants (60.2%) were males and 367(39.69%) were females. The participants suffered from major depressive disorder (422;45.8%), followed by bipolar affective disorder (392; 42.51%) and schizophrenia (108;11.7%). Most participants were in a younger age group with (80%) of population being below 50 years old and had education level below secondary education (57.4%). In the analysis of association between EQ-VAS (subjective quality of life scale) and demographic factors, a significant association was found for marital status(p<0.001), gender (p< 0.001) and

**Data Availability Statement:** Our study data source is a multi centre study conducted in Pakistan, India and Bangladesh under impact program by York university, hence our data is part

of impact program and can be shared with their consent only. The email address for the data through which York data access committee can be contacted via focal person for IMPACT program. Email address is: najma.siddiqi@york.ac.uk.

**Funding:** The authors received no specific funding for this work.

**Competing interests:** The authors have declared that no competing interests exist.

**Abbreviations:** CGI, clinical global impression scale; EQ- 5D-5l, euro quality of life index; EQVAS, Health related quality of life visual analogue scale; FDA, US, food and drug administration; ICD 10, international classification of diseases 10 edition; IMPACT, improving mental and physical health together; IOP, institute of psychiatry, Benazir Bhutto hospital, Rawalpindi; MINI, Mini neuropsychiatric interview; SMI, severe mental illnesses; WHO, world health organization.

education (p< 0.001). Women had lower EQ-VAS scores (M = 49.43±SD = 27.72) as compared to males (M = 58.81±SD = 27.1) and individuals with lower educational status also had lower mean scores. Additionally, participants who were single, divorced or widowed also had lower mean EQVAS scores. When health related quality of life was analyzed across SMI, it was lower in all SMI, but was significantly lower for depression in both subjective and objective domains of health related of quality-of-life instrument.

## Conclusion

Health related quality of life is an important outcome measure and regular assessment of both subjective and objective aspects should be incorporated in management plans of patients suffering from severe mental illnesses.

## Introduction

Severe mental illnesses are associated with a significant global disease burden, with nearly 254 million people suffering from depression, 45 million from bipolar affective disorder, and 20 million from schizophrenia, according to the World Health Organization. These illnesses result in a significant amount of morbidity, measured in disability-adjusted life years (DALYs), and are associated with shorter life expectancies and increased risk of early mortality [1].

Health related Quality of life is broadly defined by FDA as a "subjective impact of illness and its treatment on daily life and its effect on physical, emotional and social well-being" [2].

Hence quality of life is a broader concept, involving not only the subjective burden but also includes all aspects of daily living, and interaction with others. Recent studies have shown an increase in research on health-related quality of life, highlighting its significance as a primary patient outcome [3–5].

Quality of life in patients suffering from severe mental illness is lower compared to the general population. The impact on a patient's health-related quality of life is influenced by a combination. of clinical factors, environmental factors such as social support, and demographic factors [3]. Another study examining health-related quality of life in schizophrenia identified additional factors such as clinical symptoms and the presence of comorbidities [3].

Health related quality of life assessment is also routinely used recently for many other psychiatric and physical disorders and neurocognitive disorders like Alzheimer's dementia to determine impact of illness and in various psychiatric disorders like obsessive compulsive disorder and various eating disorders and various psychotic illnesses and in addition to determine the impact of illness, it is also used to determine the effect of various treatment modalities used for these illnesses [4–7].

There are various instruments available to measure health-related quality of life in patients suffering from various psychiatric illnesses, depending on the patient's clinical characteristics and sociocultural factors, including language. International studies have used EQ-5D-5L (EURO QUALILITY OF LIFE INDEX), which is a self-administered tool used to determine the burden of severe mental illnesses at both primary and tertiary care levels [8–11]. However, there is limited evidence of studies that have used EQ-5D-5L (EURO QUALILITY OF LIFE INDEX) to measure health-related quality of life in patients with severe mental illnesses in our local settings.

This study aims to use this instrument to determine health-related quality of life in our local tertiary care setting for people suffering from severe mental illnesses, to highlight their burden and determine the major sociodemographic factors that affect health-related quality of life.

## Methods

### Study design

This was descriptive cross sectional, using retrospective record view of data.

**Study Site.** This study was conducted at outpatient and inpatient units of the Institute of Psychiatry, Benazir Bhutto hospital in Rawalpindi, which is one of major mental health facilities in northern Punjab. It is affiliated with Rawalpindi Medical University and is involved in teaching and training of post graduate residents. The hospital has a 40 -bedded indoor facility and outpatient facility having a patient flow of 200patients per day. The study was conducted under the IMPACT (Improving Physical and Mental Health Together) program at the department of Research and Development, Institute of Psychiatry, Rawalpindi.

**Study population.** All patients diagnosed with severe mental illnesses (depression with psychotic features, bipolar affective disorder, schizophrenia according to ICD 10 criteria.) [12], who were 18 years of age or older and enrolled at the study site between July 2019 and March 2021 were included in the study.

### Data variables, source of data and data collection

**Source of data.** Source of data was the Qualtrics survey where data was registered and stored under IMPACT (improving physical and mental health together) program.

**Data variables and data collection.** First SMI patients were diagnosed according using the (MINI neuro psychiatric interview), and then demographic data was extracted using WHO STEPSV 3.2 (WHO STEP WISE APPROACH TO SURVEILLANCE) instrument including age, gender, income, educational status. It is a standardized tool used for collecting demographic information of the study participants and has been used in various studies. WHO STEPS includes software and supporting materials to undertake data collection such as android devices, this helps to reduce errors, any missed data and allows remote data submission. Additionally, also has benefit of reduced time and minimizes dependency on paper [13].

The primary outcome variable was health related quality of life, which was assessed using EQ-5D-5L, a self-administered questionnaire consisting of domains of mobility, self-care, usual activities, pain and discomfort, anxiety, and depression. The EQ-5D-5L further assess a patient's subjective evaluation of their health state based on visual analogue scale (EQ-5D-VAS) between worst imaginable health state and 100, best imaginable state [14].

**Analysis and statistics.** Data was entered into SPSS version 26 and the key analytic outputs included the level of impairment in health-related quality of life in patients suffering from bipolar affective disorder, schizophrenia, and depression with psychotic features. Descriptive data analysis was used to assess the proportion of patients in each domain of the health-related quality of life questionnaire and associations with various socio demographic factors was determined through chi square test. P value of <0.05 was considered statically significant.

**Ethical considerations.** The institutional research committee at the Rawalpindi medical university was contacted for permission to use data on individuals with SMI. Detailed written informed consent was taken from each participant to use their data in the study. Data was entered in a pre-designed format based on the information from the database. Names of the patients and data on personal identifies were excluded from the analysis and final presentation of study findings and all data was fully anonymized before it was analyzed. Data in hard files

were kept under lock and key, while electronic files were password protected. Access to the data for the analysis and interpretation was limited to the authorized persons such as the principal investigator.

## Results

The study included 922 SMI patients, of whom 555 participants (60.2%) were males and 367 (39.69%) were females. The participants suffered from major depressive disorder (422;45.8%), followed by bipolar affective disorder (392; 42.51%) and schizophrenia (108;11.7%). Most participants were in a younger age group with (80%) of population being below 50 years old and had education level below secondary education (57.4%). detailed demographics are shown in Table 1:

In the analysis of association between EQ-VAS (subjective quality of life scale) and demographic factors, a significant association was found for marital status(P<0.001), gender (p< 0.001) and education (p< 0.001). Women had lower EQ-VAS scores (M = 49.43±SD = 27.72) as compared to males (M = 58.81±SD = 27.1) and individuals with lower educational status also had lower mean scores. Additionally, participants who were single, divorced or widowed also had lower mean EQVAS scores as shown in Table 2.

The analysis of EQVAS across all SMI (severe mental illnesses) revealed that patients suffering from depression had lower scores across all demographic factors as compared to the patients suffering from schizophrenia and bipolar affective disorder. Additionally, people above 65 or only completing primary education, it was lowest for schizophrenia. However, people with bipolar disorder have consistently highest scores as shown in (Table 3).

**Table 1. Demographic characteristics of severe mental illness patients visiting institute of psychiatry during 2019to 2021.**

| Demographic characteristic | | Total SMI patients N (%) | Schizophrenia n (%) | Major depression With psychotic features n (%) | Bipolar affective disorder n (%) |
|---|---|---|---|---|---|
| Gender | Male | 555(60.19) | 75 (69.4) | 207 (49.1) | 273 (69.6) |
| | Female | 367(39.69) | 33(30.6) | 215(50.9) | 119 (30.4) |
| Age in years | 18–30 | 368(39.91) | 52 (48.1) | 173 (41) | 143 (36.5) |
| | 31–45 | 368(39.91) | 40 (37) | 164 (38.9) | 164 (41.8) |
| | 46–60 | 163(17.67) | 14 (13) | 73 (17.3) | 76(19.4) |
| | >**60** | 23 (24.94) | 2(1.9) | 12(2.8) | 9(2.3) |
| Education Status | No formal education | 184 (19.84) | 15(13.9) | 106(25.1) | 63(16.1) |
| | Primary education | 156(16.9) | 15(13.9) | 73(17.3) | 68(17.3) |
| | Secondary education | 190 (20.6) | 29(26.9) | 70(16.6) | 91(23.2) |
| | Higher secondary education | 392(42.5) | 49(45.4) | 173(41) | 170(43.4) |
| Monthly Income (Rs) | <20,000 | 386(39.91) | 54(50.9) | 160(38.3) | 172(44.8) |
| | 20,000–50,0000 | 405(43.92) | 40(37.7) | 198(47.4) | 167(43.5) |
| | 51,0000–100,000 | 93(10.08) | 6(5.7) | 46(11) | 41(10.7) |
| | >100,000 | 24(2.603) | 6(5.7) | 14(3.3) | 4(1) |
| Marital status | Single | 301(32.6) | 56(51.9) | 133(31.5) | 112(28.6) |
| | Married | 514(55.7) | 34(31.5) | 243(57.6) | 237(60.5) |
| | Separated/divorced /widowed | 107(11.6) | 18(16.7) | 46(10.9) | 43(11.0) |
| | Total | 922(100) | 108(100) | 422(100) | 392(100) |

**Table 2. Association of demogrpahic factors with health-related quality of life severe mental illness patients visiting institute of psychiatry during 2019 to 2021 (n = 922).**

| DEMOGRAPHIC | | EQVAS | | |
|---|---|---|---|---|
| | | n (%) | Mean and SD | P |
| **Gender** | | | | |
| | Male | 555 (60.19) | 58.81±27.49 | **<0.001** |
| | Female | 366 (39.69) | 49.43±27.72 | |
| **Age in years** | | | | |
| | 18–33 | 367(39.80) | 56.14±27.81 | **0.006** |
| | 34–49 | 368(39.91) | 57.22±27.9 | |
| | 50–65 | 163(17.67) | 48.64±27.9 | |
| | >65 | 23(24.94) | 49.57±24.8 | |
| **Education** | | | | |
| | No formal education | 183(19.84) | 44.55±26.93 | **<0.001** |
| | Primary education | 156 (16.9) | 53.72±26.66 | |
| | Secondary education | 190 (20.6) | 58.04±27.07 | |
| | Higher secondary education | 392(42.5) | 59.10±28.13 | |
| **Income** | | | | |
| | <20,000 | 368(39.91) | 55.70 ±27.34 | **0.640** |
| | 20,000–50,000 | 405(43.92) | 54.38±28.20 | |
| | 50,000–100,000 | 93(10.08) | 57.40±29.61 | |
| | >100,000 | 24(2.603) | 55.15±27.93 | |
| **Marital status** | Single | 301(32.6) | 56.5±27.55 | **<0.001** |
| | Married | 514(55.7) | 55.37±27.61 | |
| | Separated/divorced/widowed | 107(11.6) | 49.67±30.26 | |

Abb: EQ VAS: euro quality of life visual analogue scale

When asked about the health-related quality of life participants, most participants who were suffering from depression reported problems in mobility (73%), self-care (62%), activities (70%), pain (87.5%), and anxiety/ depression (90%), ranging from mild to severe. This indicates health related quality of life of participants with depression was lower as compared to bipolar affective disorder and schizophrenia, as shown in Table 4.

## Discussion

Our study has determined that health related quality of life, a major patient outcome, was low, in all population with SMI (severe mental illness). However, patients suffering from depression showed more significant results with lower mean EQVAS scores as compared to those with schizophrenia and bipolar disorder. Demographic factors such as marital status, education and gender were found to have a significant association with health-related quality of life. Females had lower (M = 49.43 ±SD 27) scores as compared to males (M 58.81 ±SD27) and people who were less educated reported lower scores. Objectively assessing health related quality of life in all domains of Eq 5d 5l, patients suffering from depression reported more problems in mobility (73%), self-care (62%) usual daily activities (70%), perception of pain (87.5%) and symptoms of anxiety /depression (90%) ranging from mild to severe in intensity as compared to patients suffering from bipolar affective disorder and schizophrenia. To improve the overall health related quality of life in SMI population, targeted interventions for patients suffering from depression should be considered.

**Table 3. Association of health-related quality of life with demographic factors according to SMI diagnosis.**

| | EQVAS SCORES | Mean and standard deviation | | |
|---|---|---|---|---|
| | Demographics | Schizophrenia | Depression with psychotic features | Bipolar affective disorder |
| GENDER | Male | 56.66±30.01 | 49.13 ±24.62 | 66.89 ±26.39 |
| | Female | 57.27±29.15 | 42.29 ± 24.47 | 60.06. ± 29.01 |
| **Age in years** | | | | |
| | 18–33 | 57.40±28.60 | 47.2. ±26.3 | 66.39 ± 25.72 |
| | 34–49 | 59.17±31.09 | 47.11. ± 23.9 | 66.85. ± 27.49 |
| | 50–65 | 48.78±29.98 | 40.3. ± 22.96 | 56.61 ± 29.79 |
| | >65 | 30±14.14 | 35.8. ± 17.68 | 72.22. ± 16.97 |
| **Education** | | | | |
| | No formal education | 45.33±32.6 | 36.5 ± 20.84. | 57.57± 29.67 |
| | Primary completed | 39.40±30.23 | 46.25. ±22.29 | 64.89 ± 26.18 |
| | Secondary education | 61.21±28.58 | 47.86 ± 24.26 | 64.86 ± 26.52 |
| | Higher education | 62.16±26.96 | 50.02 ± 26.81 | 67.46 ± 27.13 |
| **I Income** | | | | |
| | <20,000 | 56.22±27.78 | 45.2 ± 24.78 | 65.27 ±26.05 |
| | 20,000–50,000 | 58.6±29.12 | 45.2 ± 24.2 | 64.16. ± 29.02 |
| | 50,000–100,000 | 54±40.79 | 47.3 ± 27.6 | 69.09. ± 26.07 |
| | >100,000 | 46.66±36.69 | 46.6 ± 25.16 | 50. ± 28.28 |
| **Marital status** | Single | 55.23±29.06 | 49.11±25.86 | 65.91±26.12 |
| | Married | 61.18±29.20 | 45.69±23.87 | 64.43±27.73 |
| | Divorced/widowed/separated | 51.22±32.48 | 35.50±23.87 | 64.18±27.36 |

Our study findings revealed that health related quality of life is generally low in severe mental illness patients. Various other studies have been conducted to assess the health-related quality of life in this population all of which indicate lower quality of life in these patients. For example, a 2021 study determined that patients suffering from bipolar affective disorder had poor health in both physical and mental parameters which is consistent with lower health related quality of life in this SMI population [15].

Our study findings indicate that patients with depression have a lower health related quality of life than those with other severe mental health illnesses to other severe mental illnesses which is consistent with the results of a study conducted in 2020 [16]. The decreased quality of life in patients with depression could be due to clinical features of depression, such as physical symptoms of fatigue, insomnia and decreased appetite, reduced activity and mobility, sleep problems, as well as emotional and cognitive symptoms like concentration and memory problems. Studies have also shown that severity of symptoms, number of episodes and residual symptoms can impair the health-related quality of life in such patients [17].

The low socioeconomic status of majority of population in low- and middle-income countries such as Pakistan results in poor access to mental health facilities and inadequate treatment leading to lower health related quality of life. Additionally, lack of awareness about mental illness due to low educational status about disease course, prognosis and proper treatment and follow ups can be a cause of its association with lower health related quality of life. These findings are consistent with the study conducted in 2020 patients that identified age, income, and CGI scores as potential influencers on health-related quality of life which is consistent with demographic factors identified in our study, such as education, gender and age and marital status [16]

**Table 4. Frequency of reported problems according to health-related quality of life instrument (eq-5d-5l) dimensions inpatients suffering from SMIs visiting institute of psychiatry during 2019 to 2021.**

| Dimensions of EQ-5D-5L Instrument for health-related quality of life | Total SMI (N) % | Schizophrenia n (%) | Depression with psychotic features n (%) | Bipolar affective disordern (%) |
|---|---|---|---|---|
| Mobility | | | | |
| No problem | 392(100) | 56 (51.9) | 116(27.5) | 220 (56.1) |
| Slight problems | 237(100) | 23(21.3) | 126(29.9) | 88(22.4) |
| Moderate problems | 161(100) | 18(16.7) | 95 (22.5) | 48 (12.2) |
| severe problems | 121(100) | 11 (10.2) | 80(19) | 30(7.7) |
| Unable to mobilize | 11(100) | 0 (0) | 5 (1.2) | 6 (1.5) |
| Self-care. | | | | |
| No problem | 474(100) | 50 (46.3) | 159 (37.7) | 265(67.6) |
| Slight problems | 199(100) | 14(13) | 117(27.7) | 68(17.3) |
| Moderate problems | 124(100) | 23(21.3) | 70(16.6) | 31(7.9) |
| severe problems | 110(100) | 19 (17.6) | 66(15.6) | 25(6.4) |
| unable to do self-care | 15(100) | 2 (1.9) | 10 (2.4) | 3 (0.8) |
| Activity. | | | | |
| No problems | 386(100) | 29 (26.9) | 125(29.6) | 232(59.2) |
| Slight problems | 203(100) | 28 (25.9) | 106(25.1) | 69(17.6) |
| Moderate problems | 151(100) | 26(24.1) | 81(19.2) | 44(11.2) |
| Severe problems | 155(100) | 22(20.4) | 96(22.7) | 37(9.4) |
| Unable to perform activity | 27(100) | 3(2.8) | 14(3.3) | 10 (2.6) |
| Pain/ discomfort | | | | |
| No pain | 201(100) | 36(33.3) | 51(12.1) | 201(29.1) |
| Slight pain | 211(100) | 23(21.3) | 81(19.2) | 107(27.3) |
| Moderate pain | 303(100) | 30(27.8) | 167(39.6) | 106(27) |
| severe pain | 180(100) | 18(16.7) | 105(24.9) | 57(14.5) |
| extreme pain | 27(100) | 1(0.9) | 18(4.3) | 8(2) |
| Anxiety/depression. | | | | |
| Not anxious | 228(100) | 44(40.7) | 39(9.2) | 145(37.0) |
| Slightly anxious | 199(100) | 24(22.2) | 79(18.7) | 96(24.5) |
| Moderately | 252(100) | 19(17.6) | 156(37) | 77(19.6) |
| anxious | 185(100) | 18(16.7) | 110(26.1) | 57(14.5) |
| Severely | 58(100) | 3(2.8) | 38(9) | 17(4.3) |
| Extremely anxious/depressed | | | | |

Another study a study conducted under the STAR D trial revealed that demographic factors such as low education status, poor socioeconomic status, and poor social support were associated with lower health related quality of life in depression. [17] Additionally, people who are widowed or divorced or separated also exhibited lower health related quality of life. Findings of our study are consistent with this study, as indicated by lower EQVAS scores in participants who are single, divorced or widowed.

The lower health related quality of life in females can be attributed to various factors. Firstly, studies have reported general low health related quality of life reported in female population. [18]. Additionally in low- and middle-income countries like Pakistan, females have low education and socioeconomic status as compared to males making these factors more relevant in contributing to lower health quality of life in this population. Stigmatization may also play a role in this disparity.

## Strengths and limitations

Our study is one of very few studies from Pakistan that has found out the burden of severe mental illnesses on health-related quality of life using standard Eq- 5d 5l questionnaire and is also unique in identifying associated demographic factors impacting health related quality of life in a diverse population of low- and middle-income country like Pakistan.

However, we did not examine specific clinical features such as the nature and severity of symptoms and comorbidities, while other studies, such as one conducted in the United States, have identified clinical factors that contribute to health-related quality of life impairment in patients with schizophrenia. These factors include positive and negative symptoms of schizophrenia [19].

Another limitation of our study was that owing to correlational study design, it only allowed us to explore associations between demographic factors and health related quality of life without any long term or causal relationships that were beyond the scope of this study. Additionally owing to a limited sample size and participants in the study being from tertiary care setting, results cannot be generalized to entire Pakistani population.

Our study also had other limitation being not having any other tool than EQ5D5L, to compare our findings of health-related quality of life. Additionally, findings of EQVAS in our population are not compared with those without SMI.

## Conclusion

Our findings suggested that health related quality should be considered important outcome measure in management of SMI patients. Regular assessment of health-related quality of life using both objective and subjective measures should be incorporated in routine or regular checkups during follow up visits. This could lead to person centered plans that address specific needs of individual patient This approach could lead to a better person-centered care as compared to the clinician-oriented approach.

## Acknowledgments

This research was conducted through the Structured Operational Research and Training Initiative (SORT IT), a global partnership led by the Special Program for Research and Training in Tropical Diseases at the World Health Organization (WHO/TDR). The training model is based on a course developed jointly by the International Union Against Tuberculosis and Lung Disease (The Union, Paris, France) and Médecins Sans Frontières (MSF, Geneva, Switzerland). The specific SORT IT program that resulted in this publication was implemented by Common Management Unit (TB, HIV/AIDS and Malaria), through the support of the IMPACT program in collaboration with Institute of psychiatry, Rawalpindi, Pakistan.

## Author Contributions

**Conceptualization:** Zarnain Umar, Zona Tahir.

**Data curation:** Zarnain Umar, Zona Tahir.

**Formal analysis:** Zarnain Umar.

**Investigation:** Zarnain Umar.

**Methodology:** Zarnain Umar.

**Project administration:** Zarnain Umar.

**Resources:** Zarnain Umar.

**Software:** Zarnain Umar.

**Supervision:** Asad Nizami.

**Validation:** Zarnain Umar.

**Visualization:** Zarnain Umar.

**Writing – original draft:** Zarnain Umar, Zona Tahir.

**Writing – review & editing:** Zarnain Umar, Zona Tahir.

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
