## [Decision Letter · Decision Letter 0]

3 Apr 2023

PONE-D-23-01479Impact of severe mental illnesses on health-related quality of life in patients visiting institute of psychiatry, Rawalpindi during 2019 to 2021, a cross sectional studyPLOS ONE

Dear Dr. umar,

Thank you for submitting your manuscript to PLOS ONE. After careful consideration, we feel that it has merit but does not fully meet PLOS ONE’s publication criteria as it currently stands. Therefore, we invite you to submit a revised version of the manuscript that addresses the points raised during the review process.

 The Reviewers have performed valuable suggestions to improve your manuscript. Please make sure you address their concerns throughout all sections (abstract, introduction, statistical analyses, discussion).

We look forward to receiving your revised manuscript.

Kind regards,

Eleni Petkari

Academic Editor

PLOS ONE

Journal Requirements:

2. In the ethics statement in the manuscript and in the online submission form, please provide additional information about the patient records/samples used in your retrospective study. Specifically, please ensure that you have discussed whether all data/samples were fully anonymized before you accessed them and/or whether the IRB or ethics committee waived the requirement for informed consent. If patients provided informed written consent to have data/samples from their medical records used in research, please include this information.

a) The name of the colleague or the details of the professional service that edited your manuscript.

b) A copy of your manuscript showing your changes by either highlighting them or using track changes (uploaded as a *supporting information* file).

c) A clean copy of the edited manuscript (uploaded as the new *manuscript* file). 

4. Thank you for stating the following in your Competing Interests section: "no authors have competing interests"

7. Please amend your list of authors on the manuscript to ensure that each author is linked to an affiliation. Authors’ affiliations should reflect the institution where the work was done (if authors moved subsequently, you can also list the new affiliation stating “current affiliation:….” as necessary).

8. Please upload a new copy of Supplementary Figure as the detail is not clear. Please follow the link for more information:

https://blogs.plos.org/plos/2019/06/looking-good-tips-for-creating-your-plos-figures-graphics/

https://blogs.plos.org/plos/2019/06/looking-good-tips-for-creating-your-plos-figures-graphics/

Reviewers' comments:

Reviewer's Responses to Questions

**Comments to the Author**

1. Is the manuscript technically sound, and do the data support the conclusions?

Reviewer #1: Partly

Reviewer #2: Partly

Reviewer #3: Yes

2. Has the statistical analysis been performed appropriately and rigorously? 

Reviewer #1: Yes

Reviewer #2: No

Reviewer #3: Yes

3. Have the authors made all data underlying the findings in their manuscript fully available?

Reviewer #1: No

Reviewer #2: Yes

Reviewer #3: Yes

4. Is the manuscript presented in an intelligible fashion and written in standard English?

Reviewer #1: No

Reviewer #2: Yes

Reviewer #3: No

5. Review Comments to the Author

Reviewer #1: General

This is an interesting descriptive study. The analysis, while basic, is appropriate. More sophisticated analyses looking at predictors of health-related quality of life could have been undertaken, but as it stands this provides a descriptive look at health related quality of life in patients with SMI in Pakistan. In places, the sentences do not have full stops, and the phrasing is unclear. It would benefit from a careful proof-reading. I felt the conclusion in the final paragraph were not fully supported by the results of the study – without providing comparison to the general population.

• References: These are not fully formatted, some are missing titles and journal names, and some reference numbers in the text do not match the references in the reference list (e.g. reference 13 as noted below).

• Author contributions are only listed for the first author and it is therefore unclear what the other two authors contributed.

Introduction:

• Reference 2 would be better linked to where the FDA define this, given it’s a quote, rather than to the protocol for this study.

• This sentence needs to be split into the two separate points – what quality of life is, and the evidence for it. Is there a reference for recent evidence? “Hence quality of life is a broader concept, involving not only the subjective burden but also includes all aspects of daily living, interaction with others, and recent evidence and evidence on health-related quality of life is increasing and is being regarded as a major patient outcome”.

• The references are not ordered numerically in the text, instead going reference 1, 2, 3, 15, 13, 4 etc.

• Eq 5d 5l should be explained at least briefly, or spelled out in the introduction.

• Reference 13 is the ICD-10 classification, but the text refers to reference 13 as a 2018 study on health related quality of life in people with schizophrenia. Please check all the references are correctly labelled.

Methods:

• Could the authors provide a reference for WHO Steps – it wasn’t clear what this is – is it an IT solution for recording data?

• And a reference for the EQ-5D-5L?

• Under data collection could you detail how the data was collected during the study? Was this face-to-face interview, or self-completed questionnaire for example?

Results:

• P=0.000, should be written as p<0.001 as it is just a rounding up ion the statistical software and p unlikely represents 0.

• “Eq vas scores were lower in female (49.43±27.72)” Is this mean and SD? If so, they should be labelled as such.

• Table 2: Age and marital status are also associated with EQVAS in your results, but you don’t make mention of them in your results or discussion section – why is that? Would also be useful to discuss why perhaps you don’t find an association with income, as you would perhaps expect that there would be one.

• “When eq vas was analyzed across SMI (severe mental illnesses), the Eqvas score was lower for patients suffering from depression across all demographic factors as compared to the patients suffering from schizophrenia and bipolar affective (table 3)”. But there were some interesting differences – yes, in general eqvas was lowest in those with depression, but for people over age 65 or only completing primary education it was lowest in those with schizophrenia. People with bipolar disorder have consistently the highest scores though, which is interesting and worth noting.

• Is it possible to have table 1 broken down by diagnosis as a supplementary table, or numbers included in table 3? I just wonder if there are different age profiles and other demographic differences between the three diagnoses?

• Table 3: Please label as mean and SD.

• Table 3: Is there a reason there is no significance testing for this table?

• Table 4: Could you add % for the total SMI please?

• Table 4: Is “unable to” a non-response?

• I think it’s really interesting how well the bipolar disorder cohort report their quality of life: for self-care and activity particularly. Also interesting that people with schizophrenia have the highest amount of moderate/severe problems of self-care. Also interesting how many of the patients with depression have mobility issues – is there a reason this is so high? I think these points could be discussed more in the discussion.

Discussion:

• “Our study identified that health related quality of life, which is major patient outcome was lowered in all SMI (severe mental illness) population” – lower compared to what? You do not make a comparison in your study. Is there a reference “normal range”? The final conclusion also supports the need for more intervention on health related quality of life for those with SMI. While I agree, I am not sure that this study provides any evidence for that without comparing scores to a population without SMI. If there is a range or other study in those without SMI that you were comparing these results to, then you should include them to strengthen this point.

• “hence the factor of poor socio economic status becomes more relevant to contributing to lower health related quality of life in our population.” – but you didn’t find any difference by income? Could you discuss how your findings fit into this statement?

• “another factor that can be postulated is that patients suffering from depression have preserved insight in to their illness and are more aware on effect of illness on their quality of life as compared to patients suffering from bipolar affective disorder especially with psychotic features and schizophrenia” Do the authors think this explains the difference for their cohort, given they were only looking at depression with psychotic features and therefore insight might also be less in this group?

• “Another study a study conducted under STAR D trial showed that demographic factors associated with lowered health related quality of life in depression, including low education, socioeconomic status, and poor social support, and people who are widowed or divorced or separated [17]” The authors could expand on this to include their findings re marital status which is in the table but not reported in the text.

• “Another limitation of our study was that as our study design was correlational, the demographic factors that have impact on health-related quality of life were studied to determine any long term or causal relationship between them.” I am not sure what this means? Do you mean you weren’t looking for long term or causal relationships?

• Discussion of broader limitations would also be useful – how representative is this of patients in Pakistan do you think? What could have biased your findings? These are only descriptive so perhaps the different demographic characteristics in those with depression vs schizophrenia or bipolar disorder could have affected your results? How good a tool is the EQ 5D-5L? and why was this the most appropriate for the study? Are there other tools that would have strengthened the findings? Did everyone answer all the questions and could this have introduced bias? Could the way the data was collected influence how people responded?

Reviewer #2: This study has been done well. This study could have focused on understanding the Quality of life of individuals in depth instead of only profiling clients with regard to sociodemographic. Qualitative data could have enhanced the results from the survey and quantitative data which also would have given the reader key insights.

Reviewer #3: Thank you for giving me the opportunity to review this interesting article.

In general, the content of the article is appropriate but the writing should be improved.

I have carefully reviewed the content of each section and provided detailed comments. Additionally, I have taken the liberty of offering suggestions to improve the writing, which I believe are appropriate. I hope this will not be misinterpreted by the authors and they can take it as a guide.

Additionally, the authors should review the journal format and make it consistent.

TITLE

The title would be more correct in the following way:

Impact of severe mental illnesses on health-related quality of life among patients attending the Institute of Psychiatry, Rawalpindi from 2019 to 2021: a cross-sectional study.

ABSTRACT

This section should be rewritten according to the recommendations I will make throughout the paper.

Keywords should not be acronyms.

INTRODUCTION

The introduction provides a comprehensive overview of the global burden of severe mental illnesses, their impact on health-related quality of life, and the use of assessment instruments to determine the effect of various treatment modalities. It also highlights the need to study health-related quality of life in patients with severe mental illnesses in the local settings and using the EQ-5D-5L instrument. To further strengthen the introduction, it may be useful to include information on the prevalence of severe mental illnesses in Pakistan, particularly in the study location of Rawalpindi, and how this compares to global figures.

On the other hand, in terms of writing, it is very difficult to read, in addition to spelling errors. What is the reason for the authors not listing the references in order?

My suggestion for the wording is as follows.

First paragraph

I would phrase it in this way.

“Severe mental illnesses are associated with a significant global disease burden, with nearly 254 million people suffering from depression, 45 million from bipolar affective disorder, and 20 million from schizophrenia, according to the World Health Organization. These illnesses result in a significant amount of morbidity, measured in disability-adjusted life years (DALYs), and are associated with shorter life expectancies and increased risk of early mortality.”

Second paragraph

I would phrase it in this way.

“Hence quality of life is a broader concept, involving not only the subjective burden but also includes all aspects of daily living, interaction with others. Recent studies have shown an increase in research on health-related quality of life, highlighting its significance as a primary patient outcome.”

Third paragraph

I would phrase it in this way.

“Quality of life in patients suffering from severe mental illness is lower compared to the general population. The impact on a patient's health-related quality of life is influenced by a combination of clinical factors, environmental factors such as social support, and demographic factors [3]. In a cross sectional survey conducted in Germany in 2020, which included various severe mental illnesses such as dementia in addition to schizophrenia, major depressive disorder, anxiety disorders and neurotic disorders, revealed a lower health-related quality of life across all conditions. The study identified demographic factors such as age, lack of social support, and income as major predictors of quality of life impairment [15]. Another study carried out in 2018, examining health-related quality of life in schizophrenia identified additional factors such as clinical symptoms and the presence of comorbidities [13].”

Fourth paragraph

I would phrase it in this way.

“Recently, health-related quality of life assessment has become a routine practice for evaluating the impact of many psychiatric and physical disorders, as well as neurocognitive disorders such as Alzheimer's dementia [4]. This assessment is also used to evaluate various psychiatric disorders including obsessive-compulsive disorder, eating disorders, and various psychotic illnesses. Additionally, it is used to determine the effectiveness of different treatment modalities for these illnesses [5-7].”

Fifth paragraph

I would phrase it in this way.

“There are various instruments available to measure health-related quality of life in patients suffering from various psychiatric illnesses, depending on the patient's clinical characteristics and sociocultural factors, including language. International studies have used EQ-5D-5L to determine the burden of severe mental illnesses at both primary and tertiary care levels [8-12]. However, there is limited evidence of studies that have used EQ-5D-5L to measure health-related quality of life in patients with severe mental illnesses in our local settings. This study aims to use this instrument to determine health-related quality of life in our local tertiary care setting for people suffering from severe mental illnesses, in order to highlight their burden and determine the major sociodemographic factors that affect health-related quality of life.”

METHODS

Study Design

I would phrase:

“This was a descriptive cross-sectional study, using retrospective record view of data.”

Study Site

I would phrase it in this way.

“The study was conducted at the outpatient and inpatient units of the Institute of Psychiatry located at Benazir Bhutto Hospital in Rawalpindi, which is one of the major mental health facilities in northern Punjab. It is affiliated with Rawalpindi Medical University and is involved in teaching and training of postgraduate residents. The hospital has a 40-bed indoor facility and an outpatient facility with a patient flow of 200 patients per day. The study was conducted as part of the IMPACT (Improving Physical and Mental Health Together) program at the Department of Research and Development, Institute of Psychiatry, Rawalpindi.”

The information provided seems sufficient to understand the study site and its context. It mentions the name of the hospital and the university it is affiliated with, the types of units (outpatient and inpatient), the capacity of the indoor facility, the patient flow of the outpatient facility, and the fact that the study was conducted under a specific program at the department of Research and Development.

Study population

I would phrase it in this way.

“All patients diagnosed with severe mental illnesses (depression with psychotic features, bipolar affective disorder, and schizophrenia) according to the ICD-10 criteria [13], who were 18 years of age or older and enrolled at the study site between July 2019 and March 2021, were included in the study.”

It provides sufficient information about the study population.

I would divide the section "Data variables, source of data and data collection" into two different ones: Data collection method and Variables and sources of data

Data collection method

Qualtrics survey would be the data collection method

Variables and sources of data

I would phrase it in this way.

“First, SMI patients were diagnosed using the MINI neuro psychiatric interview, and then demographic data was collected using the WHO STEPS module for age, gender, income, and educational status. The primary outcome variable was health-related quality of life, which was assessed using the EQ-5D-5L questionnaire, consisting of domains such as mobility, self-care, usual activities, pain and discomfort, and anxiety and depression. The EQ-5D-5L also included a visual analogue scale (EQ-5D-VAS) to allow patients to subjectively evaluate their health state on a scale of 0 (worst imaginable health state) to 100 (best imaginable health state).”

A clearer presentation would be achieved if the authors divided the variables into independent variables (clinical and sociodemographic), and dependent variable/s.

Analysis and statistics

I would phrase it in this way.

“The data was entered into SPSS version 26, and the key analytical outputs included the level of impairment in health-related quality of life in patients with bipolar affective disorder, schizophrenia, and depression with psychotic features. Descriptive data analysis was used to assess the proportions of patients in each domain of the quality-of-life questionnaire, and associations with various sociodemographic factors were determined using a chi-square test. A p-value of <0.05 was considered statistically significant.”

The data analysis described appears to be appropriate for the study. Descriptive analysis can help provide a better understanding of the patient population and determine the proportions of patients in each domain of the quality-of-life questionnaire. The chi-square test can help identify associations between sociodemographic factors and quality-of-life outcomes in patients with severe mental illness.

Ethical considerations

I would phrase it in this way.

“The institutional research committee at Rawalpindi Medical University was contacted for permission to use data on individuals with SMI. The data was entered into a pre-designed format based on information from the database, and personal identifying information and patient names were excluded from the analysis and final presentation of study findings. Hard copies of the data were kept under lock and key, while electronic files were password protected. Access to the data for analysis and interpretation of results was limited to authorized persons, such as the principal investigator.”

RESULTS

I would phrase it in this way.

“The study included 922 SMI patients, of whom 555 (60.2%) were male and 367 (39.69%) were female. The participants suffered from major depressive disorder (422; 45.8%), bipolar affective disorder (392; 42.51%), and schizophrenia (108; 11.7%). The majority of participants were in the younger age group, with 80% of the population being below 50 years old, and had an education level below secondary education (57.4%). A detailed breakdown of the demographics can be found in Table 1.

In the analysis of associations between EQ-VAS (subjective quality of life scale) and demographic factors, a significant association was found with gender (p=0.000) and education (p=0.000). Women had lower EQ-VAS scores (49.43±27.72) compared to men (58.81±27.1), and individuals with lower education also had lower mean scores. Details can be found in Table 2.

The analysis of EQ-VAS scores across severe mental illnesses (SMI) revealed that patients with depression had lower scores across all demographic factors compared to patients with schizophrenia and bipolar affective disorder, as shown in Table 3.

When asked about their health-related quality of life, most participants who were suffering from depression reported experiencing problems with mobility (73%), self-care (62%), activities (70%), pain (87.5%), and anxiety/depression (90%), with severity ranging from mild to severe. This indicates that the health-related quality of life of participants with depression was lower compared to those with bipolar affective disorder and schizophrenia, as demonstrated in Table 4.”

Setting aside the issue of wording, this section provides a succinct summary of the study's participants and key findings, with references to the relevant tables.

To enhance comprehension, it is advised that the authors standardize the format and improve the presentation of the tables.

DISCUSSION

I would phrase it in this way.

“Our study has determined that the health-related quality of life, a major outcome for patients, was decreased in all populations with severe mental illness (SMI). However, patients suffering from depression showed more significant results with lower mean EQ-VAS scores compared to those with schizophrenia and bipolar disorder. Demographic factors, such as gender and education, were found to have a significant association with health-related quality of life. Females had lower scores (49.43 ± 27) than males (58.81 ± 27), and people with lower education levels reported lower scores. Objectively assessing health-related quality of life in all domains of EQ-5D-5L, patients with depression reported more problems with mobility (73%), self-care (62%), usual daily activities (70%), perception of pain (87.5%), and symptoms of anxiety/depression (90%), ranging from mild to severe in intensity, compared to patients with bipolar disorder and schizophrenia. To improve overall health-related quality of life for SMI populations, targeted interventions for patients with depression should be considered.

Our study revealed that severe mental illness patients have a generally low health-related quality of life. Various other studies have been conducted to assess the health-related quality of life in this population, all of which indicate a lower quality of life in these patients. For example, a 2021 study determined that patients with bipolar affective disorder had poor health in both physical and mental health parameters, which is consistent with the lower health-related quality of life found in this SMI population [14]. Similar findings have been reported in other studies, which are comparable to the results of our study. ¿References?

Our study findings indicate that patients with depression have a lower health-related quality of life than those with other severe mental illnesses, which is consistent with the results of a study conducted in 2020 [15]. The decreased quality of life in depression patients could be due to the clinical features of depression, such as physical symptoms like fatigue, insomnia, and decreased appetite, as well as emotional and cognitive symptoms like concentration and memory problems. Studies have also shown that the severity of symptoms, number of episodes, and residual symptoms can impair the health-related quality of life in these patients [17]. Additionally, patients with depression may have better insight into their illness and be more aware of its impact on their quality of life compared to those with bipolar affective disorder, particularly with psychotic features, and schizophrenia.

The low socioeconomic status of the majority of the population in low- and middle-income countries, such as Pakistan, results in poor access to mental health facilities and inadequate treatment, leading to lower health-related quality of life. Therefore, the factor of poor socioeconomic status becomes more relevant in contributing to the lower health-related quality of life in our population. Additionally, lack of awareness about mental illness due to low educational status regarding disease course, prognosis, and proper treatment and follow-ups can be a cause of its association with lower health-related quality of life. These findings are consistent with a 2020 study that identified age, income, and CGI scores as potential influencers on health-related quality of life, which is also consistent with the demographic factors identified in our study, such as education, gender, and age [15].

A study conducted under the STAR*D trial revealed that demographic factors such as low education, poor socioeconomic status, and lack of social support were associated with lower health-related quality of life in depression. Additionally, individuals who were widowed, divorced, or separated also exhibited lowered health-related quality of life [17].

The lower health-related quality of life in females may be attributed to several factors. Firstly, studies have reported lower health-related quality of life in the general female population [18]. Additionally, in low- and middle-income countries such as Pakistan, females typically have lower levels of education and socioeconomic status than males, making these factors more relevant in contributing to the lower health-related quality of life in this population. Stigmatization may also play a role in this disparity.”

Similarly, in the results section, setting aside the issue of wording, the discussion presents the study's key findings and discusses the significance of these findings in the context of previous research. It also identifies demographic factors that are associated with lower health-related quality of life in patients with severe mental illness, such as gender and education. The discussion also highlights the importance of targeted interventions for patients with depression and the need to address socioeconomic factors that contribute to lower health-related quality of life in low- and middle-income countries. Overall, the discussion provides a comprehensive summary of the study's findings and their implications.

The authors should include sections such as strengths and limitations of the study.

Strengths and limitations of the study

"Our study is among the few conducted in Pakistan that examined the impact of severe mental illnesses on health-related quality of life using the standard Eq-5d-5l questionnaire. It is also unique in identifying the sociodemographic factors that affect health-related quality of life in a diverse population of severe mental illness patients in a low- and middle-income country like Pakistan.

However, we did not examine specific clinical features such as the nature and severity of symptoms and comorbidities, while other studies, such as one conducted in the United States, have identified clinical factors that contribute to health-related quality of life impairment in patients with schizophrenia. These factors include positive and negative symptoms of schizophrenia. [16]

Another limitation was due to its correlational design, as it only allowed us to explore associations between demographic factors and health-related quality of life, without establishing any causal relationships."

Limitations that should be added:

Study that relies on retrospective record view of data is the possibility of incomplete or inaccurate data. Since the data is being gathered from past records, there is a chance that some information may be missing or incorrect.

In adittion, the study design may not allow for exploration of complex relationships or underlying factors that may influence the data, as it is limited to a descriptive analysis of the available information.

The authors should include a Conclusion section.

CONCLUSIONS

I would phrase it in this way.

“Our findings suggest that health-related quality of life should be considered an important outcome measure in the management of SMI patients. Regular assessment of health-related quality of life using both objective and subjective measures should be incorporated in routine follow-up visits. This could lead to the development of person-centered care plans that address the specific needs of individual patients, such as improved access to healthcare and social services. This approach may result in better outcomes for SMI patients, as compared to the traditional clinician-oriented approach.”

The references are not uniformly written

6. PLOS authors have the option to publish the peer review history of their article (what does this mean?). If published, this will include your full peer review and any attached files.

Reviewer #1: No

Reviewer #2: No

Reviewer #3: **Yes: **Mª Carmen Castillejos Anguiano

---

## [Author Response · Author response to Decision Letter 0]

6 May 2023

author response to reviewers.docx

---

## [Decision Letter · Decision Letter 1]

30 Jun 2023

PONE-D-23-01479R1Impact of severe mental illnesses on health-related quality of life among patients attending the Institute of Psychiatry, Rawalpindi from 2019 to 2021: a cross-sectional studyPLOS ONE

Dear Dr. umar,

Thank you for submitting your manuscript to PLOS ONE. After careful consideration, we feel that it has merit but does not fully meet PLOS ONE’s publication criteria as it currently stands. Therefore, we invite you to submit a revised version of the manuscript that addresses the points raised during the review process.

We look forward to receiving your revised manuscript.

Kind regards,

Eleni Petkari

Academic Editor

PLOS ONE

Journal Requirements:

Additional Editor Comments:

Please have the manuscript proof read by a native English speaker, as there are several unclear parts in the manuscript. See below for further requested revisions.

Reviewers' comments:

Reviewer's Responses to Questions

**Comments to the Author**

1. If the authors have adequately addressed your comments raised in a previous round of review and you feel that this manuscript is now acceptable for publication, you may indicate that here to bypass the “Comments to the Author” section, enter your conflict of interest statement in the “Confidential to Editor” section, and submit your "Accept" recommendation.

Reviewer #1: (No Response)

Reviewer #3: All comments have been addressed

2. Is the manuscript technically sound, and do the data support the conclusions?

Reviewer #1: Yes

Reviewer #3: Yes

3. Has the statistical analysis been performed appropriately and rigorously? 

Reviewer #1: Yes

Reviewer #3: Yes

4. Have the authors made all data underlying the findings in their manuscript fully available?

Reviewer #1: No

Reviewer #3: Yes

5. Is the manuscript presented in an intelligible fashion and written in standard English?

Reviewer #1: No

Reviewer #3: Yes

6. Review Comments to the Author

Reviewer #1: Thank you for the changes you have made to the manuscript. I feel that it is very much clearer, and is an interesting insight into the health related quality of life of SMI patients in Pakistan.

In places there are still missing punctuation – for example in the abstract which makes reading it hard, and the sentence structure is not always as clear as it could be. I have a couple of minor points remaining which I feel would improve the manuscript further.

Specific comments:

Introduction

“Recent studies have shown an increase in research on health-related quality of life, highlighting its significance as a primary patient outcome” – I think this sentence needs backing up, with reference to at least some recent studies or a review of recent studies.

The numbering of references is still not in numerical order, with references running 1, 2, 3, 15, 16 in the introduction.

Methods

EQ-5D-5L is spelled out in the text on the second use, rather than the first (two lines above where it is spelled out).

Could the authors add that the questionnaire was self-completed in the text for clarity?

Results

While you’ve added the p-value to the marital status in the abstract, it isn’t added to the second paragraph of the results – “p<0.001)” just needs inserting after “marital status”.

“Women had lower EQ-VAS scores (49.43±27.72) as compared to males (58.81±27.1)”: This needs labelling as (presumably) mean and standard deviation.

Table 1: Thank you for adding the results by diagnosis. I think the percentages for the total SMI patients should probably be the percentage in each group – so male: 555 (60.2), female (39.8) rather than 100 for all.

Table 2: For consistency/clarity I’d suggest reporting p-values as <0.001 rather than 0.000.

Table 2: Typo in the heading “DEMOGRPAHIC”

Discussion

“Our study has determined that health related quality of life, a major patient outcome was decreased in all population with SMI (severe mental illness).” I am still unsure about this statement – decreased compared to what? Perhaps “Our study has determined that health related quality of life, a major patient outcome was low, in all population with SMI”.

Reviewer #3: All comments have been addressed by authors.

I suggest accepting it, although there is still a need to standardize the format.

7. PLOS authors have the option to publish the peer review history of their article (what does this mean?). If published, this will include your full peer review and any attached files.

Reviewer #1: No

Reviewer #3: **Yes: **Mª Carmen Castillejos Anguiano

---

## [Author Response · Author response to Decision Letter 1]

6 Jul 2023

i have attached rebuttal letter in files section

---

## [Editor Report · Decision Letter 2]

11 Jul 2023

Impact of severe mental illnesses on health-related quality of life among patients attending the Institute of Psychiatry, Rawalpindi from 2019 to 2021: a cross-sectional study

PONE-D-23-01479R2

Dear Dr. umar,

We’re pleased to inform you that your manuscript has been judged scientifically suitable for publication and will be formally accepted for publication once it meets all outstanding technical requirements.

Kind regards,

Eleni Petkari

Academic Editor

PLOS ONE
---

## [Editor Report · Acceptance letter]

25 Jul 2023

PONE-D-23-01479R2 

Impact of severe mental illnesses on health-related quality of life among patients attending the Institute of Psychiatry, Rawalpindi from 2019 to 2021: a cross-sectional study 

Dear Dr. Umar:

I'm pleased to inform you that your manuscript has been deemed suitable for publication in PLOS ONE. Congratulations! Your manuscript is now with our production department. 

Kind regards, 

on behalf of

Dr. Eleni Petkari 

Academic Editor

PLOS ONE